# Identification and Allelopathy of Green Garlic (*Allium sativum* L.) Volatiles on Scavenging of Cucumber (*Cucumis sativus* L.) Reactive Oxygen Species

**DOI:** 10.3390/molecules24183263

**Published:** 2019-09-07

**Authors:** Fan Yang, Xiaoxue Liu, Hui Wang, Rui Deng, Hanhan Yu, Zhihui Cheng

**Affiliations:** College of Horticulture, Northwest A&F University, Taicheng Road No.3, Yangling, Shaanxi 712100, China (F.Y.) (X.L.) (H.W.) (R.D.) (H.Y.)

**Keywords:** green garlic volatile organic compounds, volatiles isolation and analysis, allelochemicals, GC-MS, biological activity, reactive oxygen species

## Abstract

Garlic and formulations containing allicin are used widely as fungicides in modern agriculture. However, limited reports are available on the allelopathic mechanism of green garlic volatile organic compounds (VOCs) and its component allelochemicals. The aim of this study was to investigate VOCs of green garlic and their effect on scavenging of reactive oxygen species (ROS) in cucumber. In this study, green garlic VOCs were collected by HS-SPME, then analyzed by GS-MS. Their biological activity were verified by bioassays. The results showed that diallyl disulfide (DADS) is the main allelochemical of green garlic VOCs and the DADS content released from green garlic is approximately 0.08 mg/g. On this basis, the allelopathic effects of green garlic VOCs in vivo and 1 mmol/L DADS on scavenging of ROS in cucumber seedlings were further studied. Green garlic VOCs and DADS both reduce superoxide anion and increase the accumulation of hydrogen peroxide of cucumber seedlings. They can also regulate active antioxidant enzymes (SOD, CAT, POD), antioxidant substances (MDA, GSH and ASA) and genes (*CscAPX*, *CsGPX*, *CsMDAR*, *CsSOD*, *CsCAT*, *CsPOD*) responding to oxidative stress in cucumber seedlings.

## 1. Introduction

Plants synthesize and release various volatile organic compounds (VOCs) [1]. VOCs play essential roles in attracting pollinators and seed-dispersers, defense against herbivores and pathogens, interplant signaling and allelopathy [2,3]. Garlic (*Allium sativum* L.), an economically important vegetable [4], contains various volatile components, including diallyl, dimethyl, and allyl methyl sulfides, disulfides, and trisulfides, as well as some other minor components, all of which are formed by the decomposition of allicin and are released upon crushing garlic. These organosulfur compounds can inhibit carcinogen activation, boost detoxifying processes, cause cell cycle arrest, stimulate the mitochondrial apoptotic pathway and increase the acetylation of histones [5]. Garlic also has potential allelopathic effects and is widely used in crop rotation and intercropping with many other crops, e.g., cucumber [6], tomato [7], eggplant [8], pepper [9]. In this kind of intercropping system, garlic has been confirmed to alleviate continuous cropping issues from the aspects of reducing plant diseases and improving the physical and chemical properties of soil [10]. Green garlic is young garlic with tender leaves that harvested at the early stage before the bulb is formed and it is consumed by people of many countries. The green garlic-cucumber intercropping system has been shown to have the benefits of improving soil fertility, promoting the activity of soil enzymes and increasing cucumber biomass [6,11]. However, there is a lack of identification of the main allelochemicals for green garlic volatiles, and knowledge about what kind of allelochemicals regulated the increased biomass of cucumber, which are the subjects in this study.

Diallyl disulfide (DADS) is a major allelochemical of the VOCs in garlic [7]. In animals, DADS has been shown having the effect of reducing cellular toxins and inhibiting the proliferation of cancer cells through several actions which include the activation of metabolizing enzymes that detoxify carcinogens, suppression of the formation of DNA adducts, antioxidant effects; regulation of cell-cycle arrest, induction of apoptosis and differentiation, histone modification and inhibition of angiogenesis and invasion [12]. Kubota et al. found that the active substances from garlic that can break the bud dormancy of grapevines are sulfur-containing compounds, among which the most effective one is DADS [13]. Previous studies found that DADS can also regulate tomato root growth by affecting cell division, endogenous phytohormone levels, expansin gene expression and the pathways of tomato root sulphate assimilation and glutathione (GSH) metabolism [7,14]. Besides, a low concentration DADS is able to promote cucumber root growth and induce main root elongation by up-regulating the expression of *CsCDKA* and *CsCDKB* genes and regulating adjusting the hormone balance of roots [15]. However, the effect of DADS on scavenging of cucumber reactive oxygen species (ROS) has not been investigated.

It is important to choose a suitable method to collect as much as possible the volatile compounds of plants in their natural state. For determination of garlic flavor components by gas chromatography-mass spectrometry (GC-MS), several sampling techniques including steam distillation (SD), simultaneous distillation and solvent extraction (SDE), microwave assisted hydrodistillation extraction (MWHD), ultrasound-assisted extraction (USE), solid-phase trapping solvent extraction (SPTE) and headspace solid-phase microextraction (HS-SPME) were applied and compared in previous studies [16,17]. Compared with SD, SDE, and SPTE, the HS-SPME method had several advantages which are rapid solvent-free extraction, no apparent thermal degradation, less laborious manipulation and less sample requirement. Five different fiber coatings were evaluated to select a suitable fiber for HS-SPME of garlic flavor components, among which the divinyl benzene/carboxen/ polydimethylsiloxane (DVB/CAR/PDMS) one was the most efficient among the investigated fibers [16]. Warren investigated the analysis of thiol compounds using a needle trap device and HS-SPME coupled to GC-MS, comparing the advantages and disadvantages of two methods [18]. Based on the above studies, the HS-SPME coupled to GC-MS method seems to be a good choice for determining the allelochemicals of green garlic. We chose a fiber-HS-SPME-GC-MS method to determine green garlic allelochemicals based on these previous studies.

Similar to other biological active factors, allelochemicals derived from plants can target some specific biological processes, which include destroying membrane permeability, influencing photosynthetic and respiratory chain electron transport, influencing cell division and ultrastructure, changing enzyme activity, altering reactive oxygen species (ROS) levels and effecting expression of related genes [19]. ROS are involved in many biological processes, such as growth, development, response of biotic and environmental stresses and programmed cell death [20,21]. It has been reported that higher ROS levels might cause oxidative damage to plants. In order to reduce this damage, plants have developed efficient antioxidant defense systems to scavenging ROS, such as superoxide dismutase (SOD), catalase (CAT), peroxidase (POD), ascorbic acid (ASA), GSH, monodehydroascorbate (MDA), the expressions of their synthetic genes are related to oxidation reactions [22,23].

In this study, we will explore the regulation mechanism of VOCs derived from green garlic in scavenging ROS of cucumber from the following three aspects. Firstly, the major effective allelochemicals of VOCs that collected from green garlic will be identified; secondly, the different ROS scavengers induced by VOCs of green garlic and the ROS levels will be investigated among treated cucumber seedlings; thirdly, the expression levels of six genes (*CsPOD, CsCAT, CscAPX, CsMDAR, CsGPX, CsSOD*) related to oxidation reactions will be checked following the treatment of cucumber seedlings with VOCs and DADS. This study will provide more information for understanding the mechanism of green garlic VOCs in scavenging ROS of vegetables.

## 2. Results

### 2.1. Identification of Volatile Allelochemicals in Green Garlic

Identification of the effective components of VOCs collected from green garlic is the basis to reveal its mechanism in scavenging ROS. For the VOCs collected from the green garlic segments that were contained in the SPME vial, only two compounds including DADS and diallyl sulfide (DAS) were identified (Figure 1A and Table 1). The retention time and relative peak area of DADS were 13.83 min and 99.52%, respectively, while, DAS had a retention time of 7.04 min and a relative peak area of 0.48%. Among the VOCs collected using green garlic segments that were put in a sealed desiccator, 17 compounds were detected (Appendix A), in which there were three sulfur compounds including DADS, methyl propenyl disulfide and DAS, with relative peak areas of 86.33%, 1.53% and 0.35%, respectively (Figure 1B and Table 1). All the other compounds were heterocyclic compounds and long chain hydrocarbon compounds, which are impurities in air. There were 42 compounds identified in the VOCs obtained from the whole green garlics that were placed in a sealed desiccator (Appendix A). Among the 42 detected compounds there was only one sulfur compound, which was DADS with a relative peak area of 15.3% (Figure 1C and Table 1), while the others were also heterocyclic compounds and long chain hydrocarbon compounds. The qualitative identification result showed that DADS is the main volatile compound of green garlic. Compared with the whole green garlic, the cut green garlic segments produced more DADS with about a 5-fold difference.

The result of an external standard method for quantitative analysis of volatiles showed that the linear relation between DADS concentration (mmol/L) (y) and peak area (x) fits a simple linear regression equation (Equation (1)):y = 5E + 09x + 3E + 07, R^2^ = 0.9903(1)

Based on this equation, the DADS that released by green garlics has a concentration of about 0.08 mg/g.

Our biological tests showed that the green garlic treatment had a consistent effect with the DADS treatment (Figure 2). When the concentration of DADS was higher than 1 mmol/L, cucumber leaves turned yellowish (Figure 2). Inversely, when the green garlic treatment was less than 2 g or the concentration of DADS was lower than 1 mmol/L, the cucumber leaves had no morphology changes (Figure 2). The amount of DADS volatilized by 2 g of green garlic approached 1 mmol/L DADS treatment based on Equation (1). In addition, higher concentrations of green garlic volatiles and DADS can cause cucumber leaves turn to soft and rot (Figure 2). These results confirmed that DADS is the main allelochemical of green garlic volatiles.

### 2.2. Effect of Green Garlic Volatiles and DADS on Cucumber ROS and Antioxidant Activity

#### 2.2.1. Effect of Green Garlic Volatiles and DADS on Cucumber ROS (O_2_^•−^ and H_2_O_2_)

Figure 3A showed the results of histological observation of H_2_O_2_ in cucumber leaves stained by DAB dye. Compared with the control groups, the colors of dyed cucumber leaves gradually turned much darker with the increasing concentrations of green garlic volatiles and DADS (Figure 3A). This suggested that green garlic volatiles and DADS led to an increasing H_2_O_2_ content at histological levels. After 10-days of co-culture with green garlic, the O_2_^•−^ contents of treated cucumber leaves were significantly lower that of control (*p* < 0.05). Similarly, the O_2_^•−^ contents of cucumber leaves under 1 mmol/L DADS treatment was also lower than that of control group (Figure 3B). The H_2_O_2_ content in cucumber leaves exhibited a rising trend with the increase of the concentration of green garlic volatiles. Besides, the H_2_O_2_ contents in cucumber leaves of all green garlic treatments and DADS were significantly higher than that of control (*p* < 0.05), which were consistent with the histological observations of H_2_O_2_ in cucumber leaves (Figure 3A).

#### 2.2.2. Effect of Green Garlic Volatiles and DADS on Cucumber Antioxidant Substances (MDA, GSH and ASA)

The MDA content of cucumber leaves co-cultured with 6-bulbs of green garlic was significantly lower than that of control (*p* < 0.05) (Figure 3D), while, that for co-cultures with 12- and 18-bulbs of green garlic the MDA content was significantly higher than the control group (*p* < 0.05) (Figure 3D). When cucumber seedlings were treated with DADS, the MDA content of cucumber leaves was also significantly lower than that of control (*p* < 0.01) which was consistent with the treatment co-cultured with 6-bulbs of green garlic (Figure 3D). The GSH content had an increasing trend with the increasing number of bulbs of green garlic. The GSH contents of 6-bulb green garlic and DADS treatment were both significantly lower than that of control (*p* < 0.05), but the treatments with 12- and 18-bulbs of green garlic were higher than that of control (Figure 3E). The ASA content in cucumbers treated with green garlic bulbs and DADS were consistently lower than that of control (Figure 3F).

#### 2.2.3. Effect of Green Garlic Volatiles and DADS on Cucumber Antioxidant Enzymes (SOD, CAT and POD)

In this study, the SOD activity of cucumber that was co-cultured with 18-bulbs of green garlic was significantly higher than other treatments (*p* < 0.05) (Figure 3G). There were no significant difference among the treatments co-cultured with 6- and 12-bulbs of green garlic, the treatment of DADS and the control groups (*p* < 0.05) (Figure 3G). All the green garlic volatiles treatments had lower CAT activity than the control (*p* < 0.01). For the CAT activity, the treatments of DADS exhibited similar results to the treatments of green garlic volatiles (Figure 3H). Both the green garlic volatiles and DADS treatments had higher POD activity than that of control (*p* < 0.01) (Figure 3I).

### 2.3. Gene Expression Changes in Response to Green Garlic Volatiles and DADS

In order to further clarify the potential mechanism of action of green garlic volatiles and DADS treatments on cucumber ROS and antioxidant enzyme activities, the expression levels of six related genes (*CscAPX, CsGPX, CsMDAR, CsSOD, CsCAT, CsPOD*) were examined with real-time quantitative polymerase chain reaction (RT-qPCR). The expression levels of these six genes were significantly changed in cucumber leaves after the treatments (Figure 4).

The green garlic volatiles and DADS treatments increased the expression levels of *CsGPX, CsMDAR*, and *CsPOD* genes and decreased the expression levels of *CscAPX, CsCAT* and *CsSOD* (Figure 4). Compared with the control group, the expression level of *CsGPX* was significantly decreased by 2-fold (*p* < 0.05) in the treatments with 12- and 18-bulbs of green garlic. The expression level of *CsPOD* in cucumber co-cultured with 12-bulbs of green garlic was increased almost 4-fold when compared with the control (*p* < 0.01) (Figure 4). However, the expression level of *CsCAT* was decreased in cucumbers that were co-cultured with green garlic, especially after 18-bulb treatment, and DADS treatment (Figure 4). The *CscAPX* expression level was decreased by green garlic volatiles and the DADS treatments (Figure 4). The expression of *CsMDAR* was increased by 2-fold when compared with the control (*p* < 0.05) (Figure 4).

## 3. Discussion

Garlic organosulfur compounds are biosynthesized for defensive purposes such as protection against abiotic stressors and are formed quickly once the plant tissues are damaged [24]. Fresh garlic only contains alliin, a derivative of cysteine. When the cells of fresh garlic cell are crushed, allinase can convert alliin to allicin. Similarly, the DADS content of the cut green garlic segments were higher than that of the whole garlic plants (Table 1), which suggested that the damaged green garlics can volatilize more DADS. In general, DATS, DADS, and VDTs are the major sulfides derived from the garlic extracts. When garlic oil is extracted by steam distillation or simultaneous distillation, the resulting extracts mainly contain DADS and DATS [25]. When garlic was treated by high temperature aging, crushing, and roasting, the compounds identified by SPME-GC/MS were also DADS and DATS [26]. DADS (97.85%), DAS (0.01%), and DATS (0.01%) were found to be the predominant flavor components of garlic samples extracted by HS-SPME using a 50/30-micron DVB/CAR/PDMS fiber [16]. In cut green garlic segments, we can detect two sulfur compounds, DADS and DAS, while in whole green garlic, only DADS can be detected (Figure 1). The green garlic allelochemicals were determined by fiber-HS-SPME-GC-MS in this study. The analytical performances of needle trap micro-extraction (NTME) coupled with GC-MS were evaluated by analyzing a mixture of twenty-two representative breath volatile organic compounds (VOCs) belonging to different chemical classes (i.e., hydrocarbons, ketones, aldehydes, aromatics and sulfurs), which confirmed the reliability of this method [27,28]. Previous studies were mainly focused on the identification of garlic compounds produced from garlic cloves, which might not be identical to those seen this study, because we collected the volatile compounds of green garlic plants

By calculation and comparison, in general, the amount of DADS released by green garlic was approximately 0.08 mg/g. The four treatments of 0, 6, 12 and 18 bulbs of green garlic in the mini-greenhouse, corresponded to the DADS concentrations of 0, 0.025, 0.046 and 0.057 μmol/L per cucumber plant (Appendix A). This means that the DADS concentration of a 6-bulb treatment in a mini-greenhouse is approximately equivalent to 50 mL of 1 mmol/L DADS treatment (0.024 μmol/L per cucumber plant). It has been reported that lower concentrations (0.01–0.62 mmol/L) of DADS can significantly promoted root growth, whereas higher levels (6.20–20.67 mmol/L) will have inhibitory effects [6]. Therefore, considering the effect and cost, 1 mmol/L DADS concentration is suitable for use.

Many studies have shown that allelochemicals significantly inhibit the activities of antioxidant enzymes, increase the level of free radicals, lead to membrane lipid peroxidation and membrane potential changes, thereby reducing the scavenging effect on ROS and destroying the entire membrane system of plants [9]. ROS play an important role in plants′ signal transduction pathways, as key regulators of processes such as growth, development, response to biotic and environmental stimuli, and plant metabolism, especially H_2_O_2_ which is important for programmed cell death to resist disease [20]. Green garlic volatiles and DADS significantly decreased the O_2_^•−^ contents (Figure 3B). However, they significantly increased the H_2_O_2_ content in cucumber leaves (Figure 3C). Plants have complex antioxidant systems to deal with ROS damage, including enzymatic systems (such as SOD, CAT and POD) and non-enzymatic systems (such as MDA, GSH and ASA) [23]. Previous studies have shown that the activities of POD, CAT, SOD and MDA contents in the leaves and roots of tomato were increased after the treatment of DADS [14]. In this study, the activity of POD was increased, while that of CAT was decreased, and the MDA content was also decreased after when treated with DADS. SOD is the most important ROS scavenger by converting O_2_^•−^ into molecular oxygen and H_2_O_2_ [23]_._ H_2_O_2_ can be decomposed into molecular oxygen and H_2_O by APX GPX, CAT, or POD [20,29]. Green garlic volatiles and DADS promoted the activities of POD and reduced the contents of GSH and ASA and the activity of CAT, then led to the change of H_2_O_2_ content (Figure 3). The gene expression changes were consistent with the changes of antioxidant enzyme and antioxidant substances. The green garlic volatiles decreased the expression of *CscAPX*, *CsGPX*, *CsCAT* and *CsPOD*. *SOD*, *CAT*, and *POD* genes play key roles in plant tissue antioxidant defenses [30]. *SOD* gene is the defense first line to against the ROS. Genes like *GPX*, *cAPX*, *MDAR*, *CAT* and *POD* work to further convert H_2_O_2_ into H_2_O and O_2_ through different reactions [23]. *GPX* utilizes glutathione (GSH) as an electron donor to reduce ROS [31]. The *POD* gene is another reported defense-related enzyme gene and 24 *POD* genes have been identified in transcriptome analysis after DADS treatment [14]. *CAT* gene directly decompose H_2_O_2_ into H_2_O and O_2_. *CAT* gene is indispensable for ROS detoxification [32]. APX enzymes catalyze the conversion of H_2_O_2_ into H_2_O and MDA using ascorbate [33]. *MDAR* helps to scavenge the MDA and generate dehydroascorbate (DHA) [34].

In summary, a scheme (Figure 5) showing green garlic volatiles′ effect on cucumber ROS, antioxidant enzymes, antioxidant substances and genes can be proposed. Firstly, green garlic volatiles decreased the expression of *CsSOD* gene, which made the O_2_^•−^ turn into molecular oxygen and H_2_O_2_. Then green garlic volatiles decreased the expression of *CscAPX, CsGPX, CsCAT* and *CsPOD* gene, which can promote the activity of POD and reduce the contents of GSH and ASA and the activity of CAT. Finally, green garlic volatiles increased the accumulation of H_2_O_2._ This increased accumulation of H_2_O_2_ might increase the disease resistance of cucumber. We will continue to work on the effects of green garlic volatiles on cucumber disease resistance in the future.

## 4. Materials and Methods

### 4.1. Materials and Equipment

Cucumber seeds of Jinyou 40, a North China fresh market type variety, were obtained from Tianjin Kernel Cucumber Research Institute (Tianjin, China). Garlic bulbs of Gailiang were provided by the College of Horticulture, Northwest A&F University (Yangling, Shaanxi Province, China). To prepare the DADS stock solutions, laboratory grade DADS (purity 80%) was purchased from Sigma-Aldrich Co. (St. Louis, MO, USA) and was firstly dissolved in Tween-80 with a ratio of 1:2 (*w*/*w*), then distilled water was added to get a 10 mmol/L stock solution which was stored at 4 °C for further use [35]. SPME fibers (50/30 µm DVB/CAR/PDMS, Sigma-Aldrich Co.) were used to absorb the VOCs and GC-MS (ISQ, Thermo Fisher, Waltham, MA, USA) was used to identify the collected VOCs. A ultraviolet spectrophotometer (UNIC 3200, UNIC, Shanghai, China) was used to measure the enzyme activities and ROS content. A microscope (BX63, Olympus, Japan) was used for histological observation of H_2_O_2_ effects.

### 4.2. Plant Growth and Experimental Design

Garlic bulbs of uniform size (approximately diameter = 4 cm, height = 4 cm and weight = 35 g) were selected and sown in plastic pots (diameter = 21 cm) filled with sterilized culture substrate. The green garlic plants that grew to 25 cm in plant height (about 30 days after sowing) were used for collecting VOCs and identification of green garlic allelopathic compounds and co-culturing with cucumber seedlings.

Germinated cucumber seeds were planted in plastic pots filled with sterilized culture substrate and cultivated in growth chamber with a day/night temperature of 25/18 °C, a day/night regime of 16/8 h and a relative humidity of 80%. The cucumber seedlings at two-true-leaf stage were used for following experiments.

To verify whether DADS is the main allelochemical of green garlic VOCs, bioassay analysis was performed. The detached cucumber leaves were treated with different weights of green garlic (0.0, 1.0, 2.0, 3.0, 5.0 g) that was cut into 1 cm segments and different concentrations of DADS (0, 0.5, 1, 2, 3, 5, and 10 mmol/L) in Petri dishes (see details in Figure 2). To ensure the treatment effect, the green garlic pieces were replaced with fresh ones and DADS was added again after 3 days of treatment. The morphology changes of these detached cucumber leaves were observed at the end of the sixth day.

To explore the effect of VOCs that derived from green garlic on scavenging ROS of cucumber, green garlic plants and cucumber seedlings were co-cultured in a greenhouse of Northwest A&F University, Yangling (N 34°16′, E 108°4′), China. Green garlic plants and cucumbers were co-cultured for 10 days in a plastic mini-greenhouse (60 × 60 × 58 cm), in which 12 cucumber seedlings were surrounded with green garlic seedlings. Four treatments including different number of green garlic seedlings (derived from 0, 6, 12 and 18 garlic bulbs, respectively) were used in this experiment (see details in Appendix A). Each treatment was replicated three times. To investigate the effect of DADS on ROS scavenging in cucumber, 12 cucumber seedlings were also planted in the plastic mini- greenhouse, and 5 mL of DADS solution with a concentration of 1 mmol/L were sprayed on each cucumber seedling, while an equal volume of distilled water was sprayed as a control (see details in Appendix A). During the 10-day cultivation of these cucumber seedlings, DADS and distilled water were sprayed twice, on the first and sixth day. For all treatments, the second true leaves of cucumber were sampled. One part of the leaves were put in a stationary liquid for histological observation of H_2_O_2_, and the other parts of the leaves were stored at −80 °C for ROS and genes assay.

### 4.3. Identification of Green Garlic Allelochemicals by Fiber-HS-SPME-GC-MS

The volatiles derived from green garlic samples were collected by three methods. For the first one, we used the headspace solid phase microextraction (HS-SPME) method, in which the green garlic plants were firstly cut into approximately 1 cm segments and 2 g samples were weighed and put into a 40 mL SPME vial (see details in Appendix A) which was pre-incubated for 10 min at 45 °C, and then the SPME fiber was exposed for 45 min at the same temperature to adsorb the volatiles. For the second and the third method, a hermetic glass container with 8 L in volume was used to collect the volatiles, the difference being that the green garlic seedlings were cut into segments or not (see details in Appendix A). Eight uniform green garlic seedlings were selected and put into these containers for three hours, then the SPME fibers were used to absorb the produced volatiles. All the collected volatile compounds were directly measured by a GC-MS system. Moreover, an external standard method was used for quantitative analysis of these volatiles, in which 2 mL DADS standard solutions with concentrations of 0.1, 0.5, 1.0, 1.5 and 2.0 mmol/L were prepared and added into a 20 mL SPME vial. Therefore, the final concentrations of DADS in SPME vials that used for the external standard method were 0.01, 0.05, 0.10, 0.15, 0.20 mmol/L, respectively.

The GC was equipped with a DB-5MS column (30 m × 0.25 mm × 0.25 μm, Agilent Technologies, Santa Clara, CA, USA). UHP helium was used as the carrier gas at a flow rate of 1 mL min^−1^. The desorption time was 2.5 min. The injector temperature was set at 230 °C and injection was performed in a split mode and split ratio of 30:1. The gas flow was set at 0.8 mL/min. The temperature was programed as follows: 40 °C for 2 min, then rising at 6 °C /min to 170 °C and held for 5 min, where it then was increased at 10 °C /min to 250 °C, where it was held for 1 min, running for 40 min. For MS, the electron multiplier was operating at 70 eV and all samples were analyzed in electron impact mode. Mass ranges of *m*/*z* 40–450 u for green garlic analysis were acquired. The temperatures of the transfer line and ion trap were both held at 250 °C. All the treatments were done in triplicates.

The compound TIC was used to identify the allelochemicals of VOCs. The Xcalibar workstation NIST standard spectrum database was used to search for each detected component, and the related literatures and standard spectra were used to check and confirm the inspection results. Only components with a matching degree greater than 800 were reported, and the relative content of each component was calculated based on the area normalization method.

### 4.4. H_2_O_2_ Histological Analysis: 3,3-Diamino-Benzidine (DAB) Staining

DAB-vascular uptake staining method was used to measure the H_2_O_2_ production. The cucumber leaves (1 cm sections) were immersed in a solution containing 1 mg/mL DAB that was dissolved in HCl-acidified (pH 3.8) distilled water, then incubated under light for 8 h, then the treated leaves were examined under a microscope [36].

### 4.5. ROS Assay

The H_2_O_2_ content in cucumber was measured according to the method described by Gong et al. using potassium iodide [37]. Superoxide anion (O_2_^•−^) content was measured according to the method of Wang et al. [38].

### 4.6. Antioxidative Enzyme and Antioxidant Substances Analysis

To assay antioxidative enzymes, 0.5 g of cucumber leaves were weighed and ground together with 6 mL of 200 mmol/L phosphate buffer (pH 7.8) which contains 1% (*w*/*v*) soluble polyvinyl pyrrolidone (PVP) under ice bath conditions. The homogenates were centrifuged at 11,000× g for 20 min at 4 °C and the supernatant were collected to measure the activities of SOD, CAT, POD and the contents of soluble protein and MDA [8]. For the assay of antioxidant substances, 1.0 g of cucumber leaves were homogenized in 5 mL of 1% (*w*/*v*) saline buffer (1% (*w*/*v*) PVP, 1 mmol/L ethylenediaminetetraacetic acid (EDTA)). Homogenates were centrifuged at 11,000 g for 15 min at 4 °C and the supernatants were collected to check the content of ASA and GSH [39].

### 4.7. Total RNA Extraction and Antioxidant Enzyme Gene Expression Analysis

Total RNA was isolated from cucumber leaves using Trizol total RNA Extraction Reagent (Bioer, Hangzhou, China). First-strand cDNA was synthesized by a HiFiScript cDNA Synthesis Ki (CoWin Biosciences, Beijing, China). The expression of selected genes was analyzed by real-time quantitative polymerase chain reaction (RTq-PCR). The corresponding primer sequences were selected from Zhao et al. [23]. The qPCR was performed with the iCycleriQTM 5 multicolor real-time PCR detection system (Bio-Rad, Hercules, CA, USA) following the manufacturer′s instructions. Gene expression levels were calculated on the basis of the 2^−ΔΔCt^ method [40]. Three biological and technical replications were performed, respectively.

### 4.8. Statistical Analysis

The statistical analyses were performed using SPSS 13.0 (IBM, Armonk, NY, USA). The significant differences between control and experimental groups that were treated with green garlic was tested by one-way ANOVA followed by Tukey′s test. The significant differences between DADS treatment and control was determined by a t-test, with * *p* values < 0.05 and ** *p* values < 0.01, respectively.

## 5. Conclusions

This study’s results showed that diallyl disulfide (DADS) is the main allelochemical of green garlic volatiles and it is released naturally at a level of approximately 0.08 mg/g green garlic. Green garlic volatiles and DADS showed obvious concentration effects. They both can reduce O_2_^−^ and they increase the accumulation of H_2_O_2_ by decreasing the expression of *CscAPX, CsGPX, CsCAT* and *CsPOD*, which may increase the disease resistance of cucumber. We will continue to study the effect of green garlic volatiles on cucumber disease resistance in the future.

## Figures and Tables

**Figure 1 molecules-24-03263-f001:**
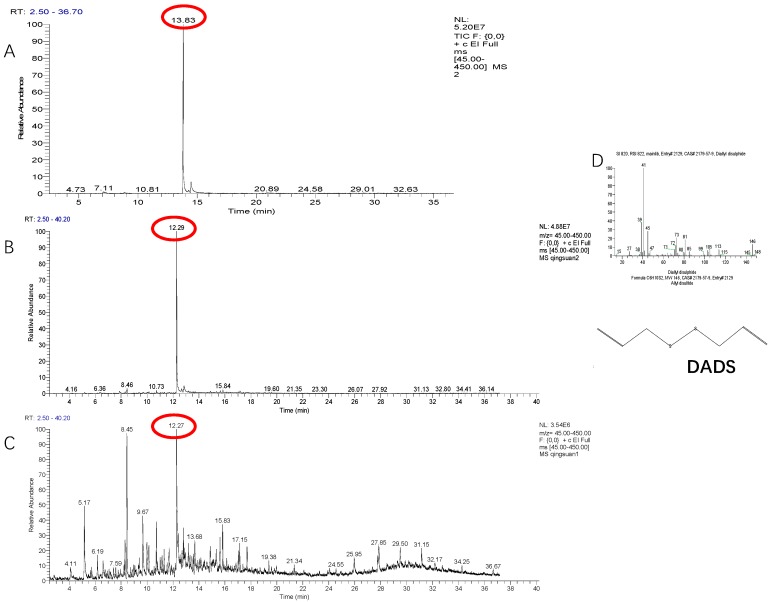
Identification of the collected volatiles from green garlic volatiles by GC-MS. (**A**) Cut (SPME vial); (**B**) Cut (sealed desiccator); (**C**) Whole (sealed desiccator); (**D**) MS spectrum and compound structure of DADS. *n* = 3, three biological replicates.

**Figure 2 molecules-24-03263-f002:**
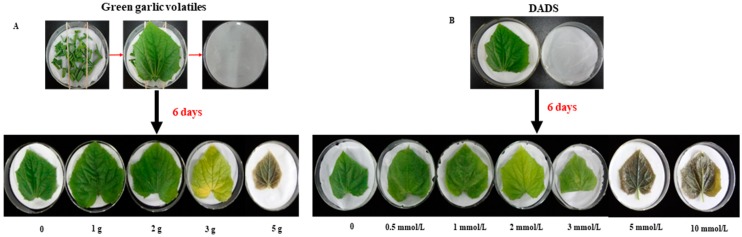
The biological test and its results of green garlic volatiles and DADS on cucumber leaf. (**A**) effect of green garlic (0.0, 1.0, 2.0, 3.0, 5.0 g) volatiles on cucumber leaf color and structure. (**B**) effect of DADS (0, 0.5, 1, 2, 3, 5, and 10 mmol/L) on cucumber leaf color and structure. Five leaves from 5 plants were used for each treatment.

**Figure 3 molecules-24-03263-f003:**
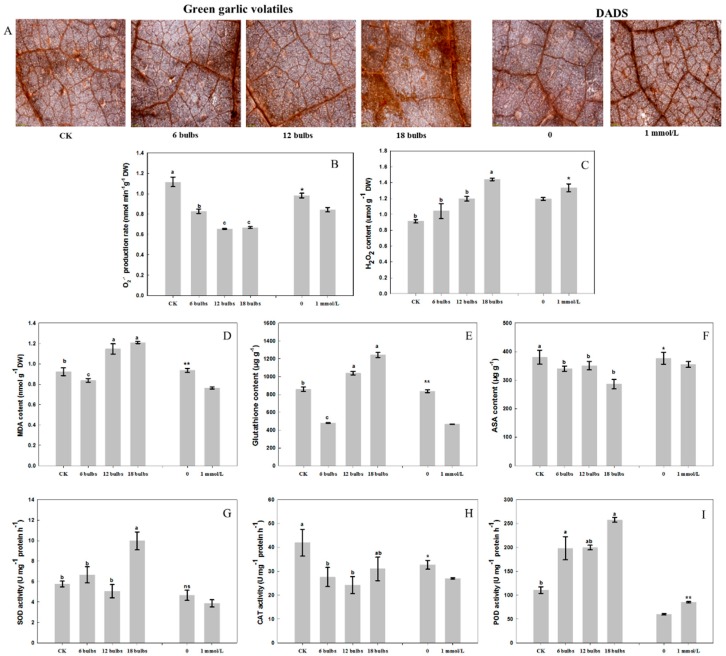
Effect of green garlic volatiles and DADS on cucumber ROS, antioxidant substance and antioxidant enzymes activity. (**A**) Histological observation of H_2_O_2_ in cucumber leaves that stained by DAB dye. (**B**,**C**), effect of green garlic volatiles and DADS on cucumber ROS (O_2_^•−^ and H_2_O_2_); (**D**–**F**), effect of green garlic volatiles and DADS on cucumber antioxidant substance content (MDA, GSH and ASA); (**G**–**I**), effect of green garlic volatiles and DADS on cucumber antioxidant enzyme activity (SOD, CAT and POD). * *p* < 0.05; ** *p* < 0.01; ANOVA, followed by Tukey test and *t*-test. Data are means ± standard errors (*n* = 3 for (**B**,**C**), three biological replicates).

**Figure 4 molecules-24-03263-f004:**
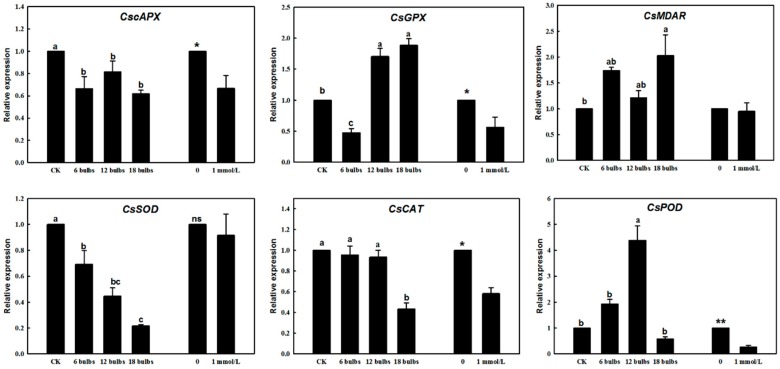
Changes in gene relative expression in response to green garlic volatiles and DADS. *CscAPX*, cucumber cytosol ascorbic acid peroxidase gene; *CsGPX*, cucumber glutathione peroxidase gene; *CsMDAR*, cucumber gene of scavenge MDA; *CsSOD*, *CsCAT* and *CsPOD*, cucumber genes of SOD, CAT, POD. * *p* < 0.05; ** *p* < 0.01; ANOVA, followed by Tukey test and *t*-test. Data are means ± standard errors (*n* = 3, three biological replicates).

**Figure 5 molecules-24-03263-f005:**
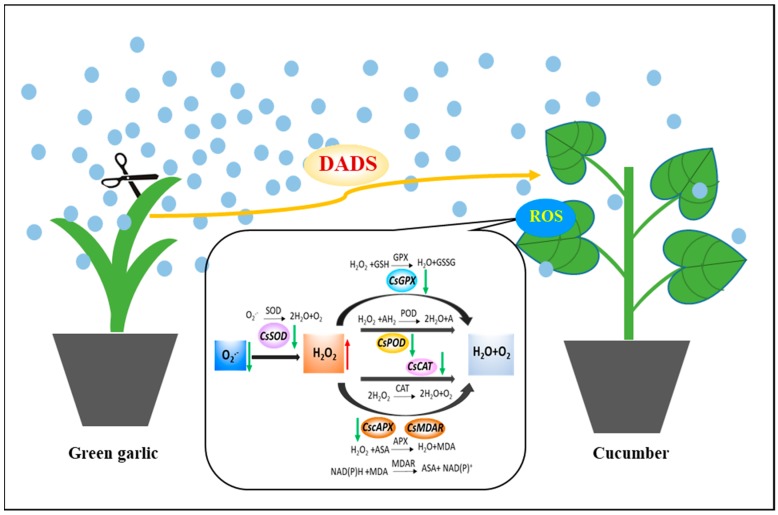
A scheme of green garlic volatiles effect on cucumber ROS, antioxidant enzymes, antioxidant substances and genes.

**Table 1 molecules-24-03263-t001:** Total sulfur compounds of green garlic volatile compounds among different collecting methods.

Treatment	Total Sulfur Compounds	1	2	3
Compound Name	Area%	RT	Compound Name	Area%	RT	Compound Name	Area%	RT
Cut (SPME vial)	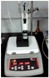 2	**Diallyl disulphide**	99.52	13.83	Methyl propenyl disulfide	0	–	Diallyl sulfide	0.48	7.04
Cut (sealed desiccator)	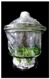 3	**Diallyl disulphide**	86.33	12.29	Methyl propenyl disulfide	1.53	7.87	Diallyl sulfide	0.35	6.36
Whole (sealed desiccator)	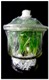 1	**Diallyl disulphide**	15.30	12.27	Methyl propenyl disulfide	0	–	Diallyl sulfide	0	–

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
