# Peer review of "Identification and Allelopathy of Green Garlic (Allium sativum L.) Volatiles on Scavenging of Cucumber (Cucumis sativus L.) Reactive Oxygen Species"

_molecules, 2019, doi:10.3390/molecules24183263_

Round 1

Reviewer 1 Report

The manuscript “Identification and Allelopathy of Green Garlic (Allium sativum L.) Volatiles on Scavenging of Cucumber (Cucumis sativus L.) Reactive Oxygen 4 Species” investigate green garlic VOCs and their effect on the plants scavenging of reactive oxygen species (ROS). The allelopathic effects of green garlic VOCs in vivo and 1 mmol/L diallyl disulfide 17 (DADS) on the cucumber seedling scavenging of ROS were further studied. Green garlic VOCs and DADS both reduce superoxide anion and increase the accumulation of hydrogen peroxide. They activate antioxidant enzymes (SOD, CAT, POD), antioxidant substances (MDA, GSH and AsA) 20 and genes (CscAPX, CsGPX, CsMDAR, CsSOD, CsCAT, CsPOD) to response to oxidative stress. The manuscript needs major revision first of all in English language and in the experimental design explanation. The design should be explained clear, simple, not in present way that is very confusing and difficult to follow. How garlic volatiles were applied to cucumber? – it should be explained simple and understandable, not in present way since it is very confusing! Was all garlic used of only its volatiles? Extensive English language and grammar corrections are necessary as well as to explain simple, clear the performed experiments and the obtained results. The overall novelty of the study should be pointed out. All abbreviations used for the first time should be explained in full. Detail comments are provided further.

Abstract: DADS – should be explained when used for first time as the abbreviation

Abstract: The statement “However, limited reports are available on green garlic volatile organic compounds(VOCs)” is not true (and should be excluded from the abstract) since relevant references on this topic were already published such as: Foods 2017, 6, 63; doi:10.3390/foods6080063; Food Sci Biotechnol (2011) 20: 775. https://doi.org/10.1007/s10068-011-0108-4; Ultrasonics Sonochemistry (2006) 13:54-60 https://doi.org/10.1016/j.ultsonch.2004.12.003; others. In addition, “In this study, green garlic VOCs were collected by HS-SPME, analyzed by GS-MS, and verified by bioassay” – the sentence is not appropriate since HS-SPME analysis is one thing and application of garlic VOCS another and it should be corrected in the mode that it is clear and exactly describes what was done in current research. Last sentence of the abstract needs rephrasing.

Page 1, line 39: “Green garlic also is a new source of allicin [12].” – thi statement is not correct since garlic is very well known as the source of allicin and there is no novelty about it to be emphasized in present manuscript.

Page 2, lines 45-49: not all used techniques were presented, e.g. ultrasonic solvent extraction is missing Ultrasonics Sonochemistry (2006) 13:54-60 https://doi.org/10.1016/j.ultsonch.2004.12.003

In distinction to very detail description of garlic biological activities, Introduction is missing data about chemical composition of garlic VOCs that were already published in different countries. Here only general data about DADs that is not enough. Therefore, detail data on garlic VOCs are important to be included. The end of Introduction is missing the important data: what is the overall novelty of the study, what was done for the first time, what is the research hypothesis and others. This is particularly important since lot of published data on garlic exist and it is necessary to point out in detail what is the overall novelty of the manuscript.

Results and discussion

Page 2, lines 72-78: why only 2 compounds were found in the headspace vial and 26 compounds in sealed desiccator of cut samples, and then 48 compounds in whole green garlic placed in desiccator? This should be explained!

Table 1 – correct IUPAC nomenclature should be used for the compound names. In addition, retention index, not retention times (that can be more or less variable) of identified compounds should be reported! Why other 26 and 48 compounds were not presented? Although they are minor constituents they should be reported I the table. Figure 1, C is clearly presented chromatographic profile of all isolated compounds that should be reported, not only the major ones.

Page 4, line 99: “DADS concentration in 2 g green garlic…” – this statement is not clear. 2g of green garlic were used, but what is the correlation with the concentration of DADS? It should be better explained.

Title 2.2.2 is not clear – what antioxidant? All paragraph need to be re-phrased because it is very difficult to follow the results and discussion!

Lines 128-129: The sentence needs re-phrasing!

Lines 25-260: “Eight uniform green garlic seedlings were put into a glass container for 3 hours to enrich volatiles.” This is not clear and should be elaborated more

Figure 3. needs better explanation and all the used abbreviation should be explained!

Figure 4. needs better explanation and all the used abbreviation should be explained!

Experimental part

Lines 164-165: “roasting by a SPME-GC/MS, the extraction…” – should be corrected, it is not clear

Paragraph 4.2 Plant growth and experimental design should be rephrased, particularly lines 235-240 and lines: 241-250. There is need to better explain experimental design and procedure applied since it is not clear. How garlic volatiles were applied to cucumber? – it should be explained simple and understandable, not in present way since it is confusing. Was all garlic applied of only its volatiles? Figure S2 does not exists, only Figure 2. However, Figure 2 is also not clearly described as “The biological tests results of green garlic volatiles (A) and DADS (B) on cucumber leaf.” There is need to explain in detail and simple the used experimental protocol and to describe Figure 2 to be understandable.

Conclusion

The conclusions have to be expanded with pointing out what new data were obtained in this study.

Author Response

Dear Editors and Reviewers:

Thank you for your letter and for the reviewers’ comments concerning our manuscript entitled “Identification and Allelopathy of Green Garlic (Allium sativum L.) Volatiles on Scavenging of Cucumber (Cucumis sativus L.) Reactive Oxygen Species” (ID: molecules-583411). Those comments are all valuable and very helpful for revising and improving our paper, as well as the important guiding significance to our researches. We have studied comments carefully and have made correction which we hope meet with approval. Revised portion are marked in red in the paper. The main corrections in the paper and the responds to the reviewer’s comments are as flowing:

Responds to the reviewer’s comments:

Response to comment: Abstract: DADS – should be explained when used for first time as the abbreviation

Response: We are very sorry for our negligence. We have explained “diallyl disulfide (DADS) ” .

Response to comment: Abstract: The statement “However, limited reports are available on green garlic volatile organic compounds(VOCs)” is not true (and should be excluded from the abstract) since relevant references on this topic were already published such as:

Foods 2017, 6, 63; doi:10.3390/foods6080063; Food Sci Biotechnol (2011) 20: 775. https://doi.org/10.1007/s10068-011-0108-4;

Ultrasonics Sonochemistry (2006) 13:54-60 https://doi.org/10.1016/j.ultsonch.2004.12.003; others.

Response: We have made correction according to the Reviewer’s comments.“However, limited reports are available on allelopathic mechanism green garlic volatile organic compounds (VOCs) and its detailed allelochemicals..”

Response to comment: In addition, “In this study, green garlic VOCs were collected by HS-SPME, analyzed by GS-MS, and verified by bioassay” – the sentence is not appropriate since HS-SPME analysis is one thing and application of garlic VOCS another and it should be corrected in the mode that it is clear and exactly describes what was done in current research.

Response: We have made correction according to the Reviewer’s comments. “In this study, green garlic VOCs were collected by HS-SPME, then analyzed by GS-MS. Their biological activity were verified by bioassay.”

Response to comment: Last sentence of the abstract needs rephrasing.

Response: We have re-written this part according to the Reviewer’s suggestion. “And they can regulate the activate antioxidant enzymes (SOD, CAT, POD), antioxidant substances (MDA, GSH and ASA) and genes (CscAPX, CsGPX, CsMDAR, CsSOD, CsCAT, CsPOD) responsing to oxidative stress in cucumber seedlings.”

Response to comment: Page 1, line 39: “Green garlic also is a new source of allicin [12].” – the statement is not correct since garlic is very well known as the source of allicin and there is no novelty about it to be emphasized in present manuscript.

Response: We have deleted the sentence according to the Reviewer’s comments.

Response to comment: Page 2, lines 45-49: not all used techniques were presented, e.g. ultrasonic solvent extraction is missing. Ultrasonics Sonochemistry (2006) 13:54-60 https://doi.org/10.1016/j.ultsonch.2004.12.003

Response: It is really true as Reviewer suggested. We have made correction according to the Reviewer’s comments. “For determination of garlic flavor components by gas chromatography-mass spectrometry (GC-MS), several sampling techniques including steam distillation (SD), simultaneous distillation and solvent extraction (SDE), microwave assisted hydrodistillation extraction (MWHD), ultrasound-assisted extraction (USE), solid-phase trapping solvent extraction (SPTE) and headspace solid-phase microextraction (HS-SPME) were performed and compared in previous studies [16, 17].”

Response to comment: In distinction to very detail description of garlic biological activities, Introduction is missing data about chemical composition of garlic VOCs that were already published in different countries. Here only general data about DADs that is not enough. Therefore, detail data on garlic VOCs are important to be included.

Response: We have added data in this part according to the Reviewer’s suggestion. “Garlic (Allium sativum L.), an important economic vegetables [4], has various volatile components include diallyl, dimethyl, and allyl methyl sulfides, disulfides, and trisulfides as well as some other minor components, all of which are formed by the decomposition of allicin and are released by crushing garlic. These organosulfur compounds can inhibit carcinogen activation, boost detoxifying processes, cause cell cycle arrest , stimulate the mitochondrial apoptotic pathway and increase the acetylation of histones [5].”

“Diallyl disulfide (DADS) is a major allelochemical of VOCs in garlic [7]. In animals, DADS has been shown having the effect of reducing cellular toxins and inhibiting proliferation of cancer cells through several actions which include the activation of metabolizing enzymes that detoxify carcinogens; suppression of the formation of DNA adducts; antioxidant effects; regulation of cell-cycle arrest; induction of apoptosis and differentiation; histone modification; and inhibition of angiogenesis and invasion [11]. Kubota et al. found that the active substances from garlic that break the bud dormancy of grapevines are sulfur-containing compounds, from which the most effective one is the DADS [12]. Previous studies found that DADS can also regulate the tomato root growth by affecting cell division, endogenous phytohormone levels, expansin genes expression and the pathway of tomato root sulphate assimilation and glutathione (GSH) metabolism [6, 13]. Besides, the low concentration DADS is able to promote cucumber root growth and induce main root elongation by up-regulating the expression of CsCDKA and CsCDKB genes and regulating adjusting the hormone balance of roots [14]. However, the eect of DADS on scavenging of cucumber reactive oxygen species (ROS) has not been investigated.”

Response to comment: The end of Introduction is missing the important data: what is the overall novelty of the study, what was done for the first time, what is the research hypothesis and others. This is particularly important since lot of published data on garlic exist and it is necessary to point out in detail what is the overall novelty of the manuscript.

Response: We have re-written this part according to the Reviewer’s suggestion. “In this study, we will explore the regulating mechanism of VOCs derived from green garlic in scavenging ROS of cucumber from the following three aspects. Firstly, the major effective allelochemicals of VOCs that collected from green garlic will be identified; Secondly, the different ROS scavengers induced by VOCs of green garlic and the ROS levels will be investigated among treated cucumber seedlings; Thirdly, the expression levels of 6 genes (CsPOD, CsCAT, CscAPX, CsMDAR, CsGPX, CsSOD) related to oxidation reactions will be checked following the treatment of VOCs and DADS to cucumber seedlings. This study will provide more information for understanding the mechanism of green garlic VOCs in scavenging ROS of vegetables.

Response to comment: Page 2, lines 72-78: why only 2 compounds were found in the headspace vial and 26 compounds in sealed desiccator of cut samples, and then 48 compounds in whole green garlic placed in desiccator? This should be explained!

Response: As Reviewer suggested that should be explained. We reconfirmed “For the VOCs collected using the green garlic segments that put in sealed desiccator, 17 compounds were detected (Table S1), in which there were three sulfur compounds including DADS, methyl propenyl disulfide and DAS with a relative peak area of 86.33%, 1.53% and 0.35%, respectively (Figure 1B and Table 1). All the other compounds were heterocyclic compounds and long chain hydrocarbon compounds, which are impurities in air. There were 42 compounds were identified for the VOCs got from the whole green garlics that placed in sealed desiccator (Table S2). Among the 42 detected compounds there was only one sulfur compound which was DADS with a relative peak area of 15.3% (Figure 1C and Table 1), while the other ones were also were heterocyclic compounds and long chain hydrocarbon compounds.” Because the volume of SPME vial is only 40 mL, there is little air in it. However, the volume of desiccator is 8 L, there is much air in it.

Response to comment: Table 1 – correct IUPAC nomenclature should be used for the compound names. In addition, retention index, not retention times (that can be more or less variable) of identified compounds should be reported! Why other 26 and 48 compounds were not presented? Although they are minor constituents they should be reported I the table. Figure 1, C is clearly presented chromatographic profile of all isolated compounds that should be reported, not only the major ones.

Response: We have made correction according to the Reviewer’s comments. Other compounds were presented in supporting information. “disulfide, methyl 2-propenyl” was corrected as “methyl propenyl disulfide”. The retention index need linear retention indices (RI) of n-paraffin as external references. I'm sorry we didn't do this experiment. We added the Table S1 and Table S2 to list other compounds in supporting information according to the Reviewer’s comments.

Response to comment: Page 4, line 99: “DADS concentration in 2 g green garlic…” – this statement is not clear. 2g of green garlic were used, but what is the correlation with the concentration of DADS? It should be better explained.

Response: We have made correction according to the Reviewer’s comments. “DADS volatilized by 2 g green garlic was approach to 1 mmol/L DADS treatment based on the Linear equation 1..”

Response to comment: Title 2.2.2 is not clear – what antioxidant? All paragraph need to be re-phrased because it is very difficult to follow the results and discussion!

Response: We have re-written this part according to the Reviewer’s suggestion. “2.2.2 Effect of green garlic volatiles and DADS on cucumber antioxidant substance (MDA, GSH and ASA)

The MDA content of cucumber leaves co-cultured with 6-bulb green garlics was significantly lower than that of control (p< 0.05) (Figure 3D). While, for that co-cultured with 12-and 18-bulb green garlics the MDA content was significantly higher than the control group (p< 0.05) (Figure 3D). When cucumber seedlings treated with DADS, the MDA content of cucumber leaves was also significantly lower than that of control (p< 0.01) which was consistent with the treatment that co-cultured with 6-bulb green garlics (Figure 3D). The GSH content had an increasing trend with the increasing number of bulbs of green garlics. GSH contents of 6 bulb green garlic and DADS treatment were both significantly lower than tha of control (p< 0.05), but the treatments of 12- and 18-bulb green garlics were higher than that of control (Figure 3E). The ASA content in cucumbers that treated with green garlics and DADS were consistently lower than that of control(Figure 3F).

Response to comment: Lines 128-129: The sentence needs re-phrasing!

Response: We have re-written this part according to the Reviewer’s suggestion. “In this study, SOD activity of cucumber that co-cultured with 18-bulb green garlics was significantly higher than other treatments (p< 0.05) (Figure 3G)..

Response to comment: Lines 25-260: “Eight uniform green garlic seedlings were put into a glass container for 3 hours to enrich volatiles.” This is not clear and should be elaborated more.

Response: We have made explanation according to the Reviewer’s comments. “For the second and the third method, a hermetic glass container with 8 L in volume was used to collect the volatiles, while the difference is that the green garlic seedlings were cut into segments or not (see details in Figure S2). Eight uniform green garlic seedlings were selected and put into these containers for three hours, then the SPME fibers were used to absorb the produced volatiles. All the collected volatile compounds were directly measured by GC-MS system.

Response to comment: Figure 3. needs better explanation and all the used abbreviation should be explained!

Response: We have made explanation according to the Reviewer’s comments. “Figure 3. Effect of green garlic volatiles and DADS on cucumber ROS, antioxidant substance and antioxidant enzymes activity. A, histological observation of H2O2 in cucumber leaves that stained by DAB dye. B and C, effect of green garlic volatiles and DADS on cucumber ROS (O2•− and H2O2); D, E and F, effect of green garlic volatiles and DADS on cucumber antioxidant substance content (MDA, GSH and ASA); G, H and I, effect of green garlic volatiles and DADS on cucumber antioxidant enzyme activity (SOD, CAT and POD). * p < 0.05; ** p < 0.01; ANOVA, followed by Tukey test and t-test. Data are means ± standard errors (n = 3 for B-C, three biological replicates).

Response to comment: Figure 4. needs better explanation and all the used abbreviation should be explained!

Response: We have made explanation according to the Reviewer’s comments. “Figure 4. Changes in gene relative expression in response to green garlic volatiles and DADS. CscAPX, cucumber cytosol ascorbic acid peroxidase gene; CsGPX, cucumber glutathione peroxidase gene; CsMDAR, cucumber gene of scavenge MDA; CsSOD, CsCAT and CsPOD , cucumber genes of SOD, CAT, POD. p < 0.05; ** p < 0.01; ANOVA, followed by Tukey test and t-test. Data are means ± standard errors (n = 3, three biological replicates).

Response to comment: Lines 164-165: “roasting by a SPME-GC/MS, the extraction…” – should be corrected, it is not clear

Response: We have made correction according to the Reviewer’s comments. “When garlics treated by high temperature aging, crushing, and roasting, the compounds that identified by SPME-GC/MS were also DADS and DATS [26].

Response to comment: Paragraph 4.2 Plant growth and experimental design should be rephrased, particularly lines 235-240 and lines: 241-250. There is need to better explain experimental design and procedure applied since it is not clear. How garlic volatiles were applied to cucumber? – it should be explained simple and understandable, not in present way since it is confusing. Was all garlic applied of only its volatiles? Figure S2 does not exists, only Figure 2. However, Figure 2 is also not clearly described as “The biological tests results of green garlic volatiles (A) and DADS (B) on cucumber leaf.” There is need to explain in detail and simple the used experimental protocol and to describe Figure 2 to be understandable.

Response: We have re-written this part according to the Reviewer’s suggestion. “To verify whether DADS is the main allelochemical of green garlic VOCs, the bioassay analysis was performed. The detached cucumber leaves were treated with different-weight green garlics (0.0, 1.0, 2.0, 3.0, 5.0 g) that cut into 1 cm segments and different-concentration DADS (0, 0.5, 1, 2, 3, 5, and 10 mmol/L) in petri dishes (see details in Figure 2). To ensure the treatment effect, the green garlics were replaced with fresh ones and the DADS were added again after 3 days treatment. The morphology changes of these detached cucumber leaves were observed at the end of the sixth day.

To explore the effect of VOCs that derived from green garlic on scavenging ROS of cucumber, green garlics and cucumber seedlings were co-cultured in a greenhouse of Northwest A&F University, Yangling (N 34o16′, E 108o4′), China. Green garlics and cucumbers were co-cultured for 10 days in a plastic mini-greenhouse (60 × 60 × 58 cm), in which 12 cucumber seedlings were surrounded with green garlic seedlings. Four treatments including different number of green garlic seedlings (derived from 0, 6, 12 and 18 garlic bulbs, respectively) were set in this experiment (see details in Figure S1). Each treatment was replicated three times. To investigate the effect of DADS on scavenging ROS of cucumber, 12 cucumber seedlings were also planted in the plastic mini-greenhouse, and 5 mL DADS solutions with the concentration of 1mmol/L were sprayed for each cucumber seedling, while the distilled water of equal volume was sprayed as a control (see details in Figure S1). During the 10-day cultivation of these cucumber seedlings, DADS and distilled water were sprayed twice which did on the first and sixth day. For all treatments, the second true leaves of cucumber were sampled. One part leaves were put into stationary liquid for histological observation of H2O2, and the other part leaves were stored in -80oC for ROS and genes assay.

And we corrected the Figure 2.

Response to comment: The conclusions have to be expanded with pointing out what new data were obtained in this study.

Response: We have re-written this part according to the Reviewer’s suggestion. “This study results showed diallyl disulfide (DADS) is the main allelochemical of green garlic volatiles and it is released naturally approximate 0.08 mg/g green garlic. Green garlic volatiles and DADS showed obvious concentration effect. They both can reduce O2•−. And they increase the accumulation of H2O2 by decreased the expression of CscAPX, CsGPX, CsCAT and CsPOD, it maybe increase disease resistance of cucumber. We will continue study the effect of green garlic volatiles on cucumber disease resistance in the future.”

Special thanks to you for your good comments.

We tried our best to improve the manuscript and made some changes in the manuscript. These changes will not influence the content and framework of the paper. And here we did not list the changes but marked in red in revised paper.

We appreciate for Editors/Reviewers’ warm work earnestly, and hope that the correction will meet with approval.

Once again, thank you very much for your comments and suggestions。

Thank you and best regards.

Yours sincerely,

Zhihui Cheng

Corresponding author: Zhihui Cheng

E-mail: [email protected].

Reviewer 2 Report

COMMENTS TO THE AUTHOR(S)

The manuscript is well written and well organized. In my opinion, this paper should be published in Molecules after minor revisions. I suggest to include all the analytical Figures of Merit of the HS-SPME method.

Specific comments:

L12, please include a space before the round bracket.

L15, please explain the acronym DADS.

L31, I suggest to modify the sentence as follow: … with many other crops, e.g. cucumber, tomato, eggplant and pepper.

L54, please modify the word “identify” with “determine”.

L251-275, did the authors carried out tests in order to optimize the HS analysis? If yes, please include these experimental data. In addition, it is well known that the stability of the sulfur analytes is critical, did the authors evaluated this aspect? The following articles can be useful for the authors and used for the discussion:

doi.10.1088/1752-7155/9/4/047110

doi.10.1088/1752-7163/aa94e7

L72, how the authors identified the compounds? Did they used NIST library or standard solution? In addition, did the authors used TIC or EIC? Please, explain this point.

L84, did the authors performed an home-made validation of the HS-SPME method? If yes, please include all the experimental data. Please, focus on analyte recovery and reproducibility.

L90, please modify the figure caption in order to better explain the figure. In particular, please include the MS spectrum of DADS.

L117.126, please include all the experimental values in the paragraph.

L137, please include the number of replicates and explain the error bars.

L155, please include the number of replicates and explain the error bars.

Author Response

Dear Editors and Reviewers:

Thank you for your letter and for the reviewers’ comments concerning our manuscript entitled “Identification and Allelopathy of Green Garlic (Allium sativum L.) Volatiles on Scavenging of Cucumber (Cucumis sativus L.) Reactive Oxygen Species” (ID: molecules-583411). Those comments are all valuable and very helpful for revising and improving our paper, as well as the important guiding significance to our researches. We have studied comments carefully and have made correction which we hope meet with approval. Revised portion are marked in red in the paper. The main corrections in the paper and the responds to the reviewer’s comments are as flowing:

Responds to the reviewer’s comments:

Response to comment: L12, please include a space before the round bracket.

Response: We are very sorry for our negligence. We have made correction according to the Reviewer’s comments. “green garlic volatile organic compounds (VOCs)”.

Response to comment: L15, please explain the acronym DADS.

Response: We are very sorry for our negligence. We have explained “diallyl disulfide (DADS) ” .

Response to comment: L31, I suggest to modify the sentence as follow: … with many other crops, e.g. cucumber, tomato, eggplant and pepper.

Response: We have made correction according to the Reviewer’s comments. “Garlic also has potential allelopathic effects and widely used in crop rotation and intercropping with many other crops, e.g. cucumber [6], tomato [7], eggplant [8], pepper [9].

Response to comment: L54, please modify the word “identify” with “determine”.

Response: We have made correction according to the Reviewer’s comments. “We chose Fiber-HS-SPME-GC-MS method to determine green garlic allelochemicals based on these previous studies.”

Response to comment: L251-275, did the authors carried out tests in order to optimize the HS analysis? If yes, please include these experimental data. In addition, it is well known that the stability of the sulfur analytes is critical, did the authors evaluated this aspect? The following articles can be useful for the authors and used for the discussion:

doi.10.1088/1752-7155/9/4/047110

doi.10.1088/1752-7163/aa94e7

Response: We have made explanation according to the Reviewer’s comments. Our method is the result of many years of exploration and continuous optimization. Relevant articles have been published in Chinese journals, and it is a mature and reliable method.

DOI: 10.7506/spkx1002-6630-201422039

DOI: 10.3969/j.issn.1005-6521.2014.19.025

DOI: 10.3969/j.issn.1000-9973.2013.07.026

DOI: 10.3969/j.issn.1000-6346.2013.10.013

As Reviewer suggested, the two articles are useful for us and we used them for the discussion.

Response to comment: L72, how the authors identified the compounds? Did they used NIST library or standard solution? In addition, did the authors used TIC or EIC? Please, explain this point.

Response: We have added data in this part according to the Reviewer’s suggestion. “The compound TIC was used to identify the allelochemicals of VOCs. The random Xcalibar workstation NIST standard spectrum database was used to search for each detected component, and the related literatures and standard spectrum were used to check and confirm the inspection results. Only component with matched-degree greater than 800 were reported, and the relative content of each component was calculated based on the area normalization method.

Response to comment: L84, did the authors performed an home-made validation of the HS-SPME method? If yes, please include all the experimental data. Please, focus on analyte recovery and reproducibility.

Response: We repeated three times for the results. Its reproducibility is good. Regretfully, we not did the recovery experiment.

Response to comment: L90, please modify the figure caption in order to better explain the figure. In particular, please include the MS spectrum of DADS.

Response: We have made explanation according to the Reviewer’s comments. “Figure 1. Collection and identification of main allelochemicals of green garlic volatiles by GC-MS. A, Cut (SPME vial); B, Cut (sealed desiccator); C, Whole (sealed desiccator); D, MS spectrum and compound structure of DADS. n = 3, three biological replicates.

Response to comment: L117.126, please include all the experimental values in the paragraph.

Response: We have made correction according to the Reviewer’s comments. “2.2.2 Effect of green garlic volatiles and DADS on cucumber antioxidant substance (MDA, GSH and ASA)

The MDA content of cucumber leaves co-cultured with 6-bulb green garlics was significantly lower than that of control (p< 0.05) (Figure 3D). While, for that co-cultured with 12-and 18-bulb green garlics the MDA content was significantly higher than the control group (p< 0.05) (Figure 3D). When cucumber seedlings treated with DADS, the MDA content of cucumber leaves was also significantly lower than that of control (p< 0.01) which was consistent with the treatment that co-cultured with 6-bulb green garlics (Figure 3D). The GSH content had an increasing trend with the increasing number of bulbs of green garlics. GSH contents of 6 bulb green garlic and DADS treatment were both significantly lower than tha of control (p< 0.05), but the treatments of 12- and 18-bulb green garlics were higher than that of control (Figure 3E). The ASA content in cucumbers that treated with green garlics and DADS were consistently lower than that of control(Figure 3F).

Response to comment: L137, please include the number of replicates and explain the error bars.

Response: We have made explanation according to the Reviewer’s comments. “Figure 3. Effect of green garlic volatiles and DADS on cucumber ROS, antioxidant substance and antioxidant enzymes activity. A, histological observation of H2O2 in cucumber leaves that stained by DAB dye. B and C, effect of green garlic volatiles and DADS on cucumber ROS (O2•− and H2O2); D, E and F, effect of green garlic volatiles and DADS on cucumber antioxidant substance content (MDA, GSH and ASA); G, H and I, effect of green garlic volatiles and DADS on cucumber antioxidant enzyme activity (SOD, CAT and POD). * p < 0.05; ** p < 0.01; ANOVA, followed by Tukey test and t-test. Data are means ± standard errors (n = 3 for B-C, three biological replicates).

Response to comment: L155, please include the number of replicates and explain the error bars.

Response: We have made explanation according to the Reviewer’s comments. “Figure 4. Changes in gene relative expression in response to green garlic volatiles and DADS. CscAPX, cucumber cytosol ascorbic acid peroxidase gene; CsGPX, cucumber glutathione peroxidase gene; CsMDAR, cucumber gene of scavenge MDA; CsSOD, CsCAT and CsPOD , cucumber genes of SOD, CAT, POD. p < 0.05; ** p < 0.01; ANOVA, followed by Tukey test and t-test. Data are means ± standard errors (n = 3, three biological replicates).

Special thanks to you for your good comments.

We tried our best to improve the manuscript and made some changes in the manuscript. These changes will not influence the content and framework of the paper. And here we did not list the changes but marked in red in revised paper.

We appreciate for Editors/Reviewers’ warm work earnestly, and hope that the correction will meet with approval.

Once again, thank you very much for your comments and suggestions.

Thank you and best regards.

Yours sincerely,

Zhihui Cheng

Corresponding author: Zhihui Cheng

E-mail: [email protected].

Round 2

Reviewer 1 Report

The manuscript was improved. However there is still need to improve the English grammar throughout the manuscript.